# Evaluation of the Safety and Antiproliferative Activity of Bulgarian Rose Essential Oil: An In Vitro and In Silico Model of Colorectal Adenocarcinoma

**DOI:** 10.3390/cimb47080649

**Published:** 2025-08-13

**Authors:** Rayna Nenova, Kalin Kalinov, Deyana Nedeva, Ana Dobreva, Neli Vilhelmova-Ilieva, Ani Georgieva, Ivan Iliev

**Affiliations:** 1Institute of Roses and Aromatic Plants, Agricultural Academy, 49 Osvobozdhenie Blvd., 6100 Kazanlak, Bulgaria; rayna.nenova@gmail.com (R.N.); dr.kkalinov@gmail.com (K.K.); deyana077@gmail.com (D.N.); anadobreva@abv.bg (A.D.); 2The Stephan Angeloff Institute of Microbiology—Bulgarian Academy of Sciences, Acad. G. Bonchev Str., bl. 26, 1113 Sofia, Bulgaria; 3Institute of Experimental Morphology, Pathology and Anthropology with Museum, Bulgarian Academy of Sciences, Acad. G. Bonchev Str., bl. 25, 1113 Sofia, Bulgaria; georgieva_any@abv.bg; 4Biotechnology Department, University of Chemical Technology and Metallurgy, 8 Kliment Ohridski Blvd., 1797 Sofia, Bulgaria

**Keywords:** Rosa damascena Mill essential oil, cytotoxicity, antiproliferative activity, HCT-8 and HT-29 cell lines

## Abstract

The side effects of conventional cancer treatments, such as chemotherapy, radiotherapy, etc., worsen the quality of life of patients. Therefore, it is necessary to explore the possibilities of creating new drugs containing natural products with low toxicity. The experimental scientific pharmacological research of rose preparations in Bulgaria began in the first half of the 20th century. Bulgarian rose essential oil (BREO) is qualified by GC FID analysis. To study the effect of the BREO, we used HCT-8 and HT-29 tumor cell lines. As a model of healthy tissue, we used the non-tumorigenic cells MCF-12F. Cells were treated with twofold increasing concentrations of BREO from 7.5 µg/mL to 1000 µg/mL. The NRU test and MTT assay were used for evaluation of the safety, antiproliferative activity and colony formation assay. Our results showed low cytotoxicity (CC_50_ = 629.72 ± 22.38 μg/mL) and high level of photosafety (PIF = 0.92) of BREO. The antiproliferative activity test shows that the BREO has an IC_50_ = 290.45 ± 10.79 μg/mL for the HT-29 cells. In the normal cell line MCF-12F, this effect is lower (IC_50_ = 383.90 ± 34.75 μg/mL). Furthermore, colony forming assay showed a significant reduction in IC_50_ value (IC_50_ = 163.79 ± 10.25 μg/mL) in HT-29 cells. The in silico experiments confirmed the potential of the BREO for antiproliferative effect and further activation of different pathways leading to apoptosis.

## 1. Introduction

Rosa damascena Mill essential oil is among the priciest etheric oils available worldwide [1] because of its low distillation yield and dearth of both natural and synthetic alternatives. Bulgarian rose essential oil (BREO) is an essential component of almost all high-end fragrances created by the most renowned manufacturers, including Chanel, Givenchy, Lancôme, Christian Dior and Bulgari. Aside from its extensive usage in the perfumery, cosmetics and food industries, its medicinal properties are by far its greatest benefit to humanity.

The experimental scientific pharmacological research of BREO began in the first half of the 20th century. The founders of ethnopharmacology and pharmacotherapy in Bulgaria—Professor Vladimir Aleksiev and Professor Petur Nikolov—devoted their research to the Bulgarian medicinal plants based on the experience of Bulgarian folk medicine. In the pre-antibiotic era, Prof. Aleksiev managed to achieve remarkable results with “Rosalipchin”—a rose preparation, in the treatment of lung abscesses and pulmonary tuberculosis. Based on lipchin (a quinine preparation), by adding BREO, colloidal sulfur, iodine, eucalyptus oil and menthol, a series of original preparations was created: Rosalipchin, Thiolipchin, Rosathiolipchin, Pulmochin, etc. The most popular of them is Pulmochin, which has reached us today (last included in the official “Medicinal Guide” in 1982). In his 1941 essay “The Rose as a Medicine,” Prof. Nikolov detailed his research on the oil-bearing rose—*Rosa damascena*. He offered a whole range of independent and combined preparations with anti-inflammatory, laxative and antihelmintic effects based on these studies. Later, during the 1970s, academician A. Maleev [1,2,3,4,5] continued the research into attar of roses and its pharmacology, toxicology, chemotherapeutic, pharmacokinetics and clinical effects. In the essay “Rose, rose oil, gyrosital − antisclerotic, antispasmodic, hepatoprotective, general stimulant drug.”, M. Kirov described the properties of another rose oil preparation [6]. Contributions to the biological characterization of BREO were also made by G. Neshev [7,8] and other Bulgarian, as well as foreign researchers.

Nowadays, we turn our attention to anticancer properties of Rose essential oil (REO) as a natural therapeutic agent. The antitumor effect of REO has been studied in an in vitro model of colon cancer [9,10]. It is known that normal cell growth and proliferation are regulated by tumour suppressor genes and proto-oncogenes, which control the cell cycle and apoptosis. The simultaneous activation of important oncogenes through mutation can deactivate a number of tumor suppressor genes, leading to the increased and uncontrolled proliferation of cancer cells and the advancement of colon cancer [11]. The stages of development of colorectal cancer can be categorized as initiation, promotion and metastasis. Cell proliferation in normal cells is regulated at the cell cycle checkpoints G1/S and G2/M, at which time correct DNA synthesis and chromosomal segregation are maintained [12]. Cell proliferation is important in the development of colon cancer from the initiation stage. As the cells continue to divide, they acquire more defects in key genes, allowing the cell to continue growing. Excessive generation of reactive oxygen species (ROS) might result in free radical-induced DNA damage [13]. If not rectified by the endogenous DNA repair systems, this damage may be integrated as a lasting sequence alteration, potentially triggering the multi-step processes associated with carcinogenesis.

The activation of mitogen-activated protein kinase p-38 alfa (MAPKP-38α) by ROS in cancer cells leads to apoptosis [14]. MAPKP-38α regulates cellular functions like division, stress response, differentiation, survival and immune response [15]. Despite its role in cancer suppression, numerous studies have demonstrated its involvement in cancer progression [16,17]. Dysfunctions due to mutations within the MAPK pathways have been correlated to Alzheimer’s disease, Parkinson’s disease, inflammation and multiple types of cancer [18,19,20].

A deepened insight into the mechanisms of MAPK signaling is of great interest when it comes to understanding the pathology, and to developing new therapeutics. The p38 MAPK is a signaling pathway important for cells to respond to environmental and intracellular stress. Upon activation, the p38 kinase phosphorylates downstream effectors, which control the inflammatory response and coordinate fundamental cellular processes such as proliferation, apoptosis and differentiation [21]. Dysregulation of this signaling pathway has been linked to inflammatory diseases and cancer. Secretion of glucocorticoids (GCs) is a classical endocrine response to stress. The glucocorticoid receptor (GR) is the primary effector of GCs and plays an important role in the regulation of cell metabolism and immune response by influencing gene expression in response to hormone-dependent activation. Its ligands, the GCs or steroids, are used as standard therapy for anti-inflammatory treatment, severe asthma, autoimmune diseases and several types of cancer. The GR is identified as one of the downstream targets of p38, and, at the same time, it was shown that glucocorticoids could influence p38 signaling. Corticosteroids have been exploited for their antiproliferative and antiangiogenic actions for the treatment of cancers [22,23]. Many of the biological activities of REO have been attributed to major compounds but their direct interaction with colonic epithelial cells and their genoprotective effects are not well established. In their study Thapa and colleagues had shown the potential of geraniol and other components of different essential oils to protect the colonic epithelium against methylating DNA damage. Several studies on etheric oils suggest that these natural plant secondary metabolites have therapeutic potential against colonic diseases [24,25].

The effect of REO in in vitro model of colorectal cancer may be due to the biological activities of its compounds. For example, polyphenols (gallic acid, quercetin, rutin, kaempferol) and terpenoids (citronellol, geraniol, nerol, linalool, β-phenylethyl alcohol) can function like chemopreventive and chemotherapeutic drugs, which are used to reduce the risk of developing cancer, as well as inhibit tumor growth [26]. They work by targeting specific mechanisms that contribute to the development of cancer. Such drugs are aromatase inhibitors (tamoxifen and raloxifene) and non-steroidal anti-inflammatory drugs (statins, metformin and retinoids). In search of new target-specific substances, phytochemicals have been used for the treatment of lung cancer to overcome the limitations of known chemotherapeutic drugs, such as non-specificity, resistance and toxicity [27]. Moreover, a number of studies have shown the ability of chemopreventive phytochemicals to increase the sensitivity of cancer cells to conventional anticancer drugs [28]. The final important stage in cancer development is the spread of cancer to other tissues, the process known as metastasis. Treatment for metastatic cancer often involves a combination of therapies, including surgery, chemotherapy, radiation therapy and immunotherapy. Unfortunately, conventional cancer treatments cause a variety of side effects. Phytochemicals arise as potential alternative or supplementary anti-cancer therapeutics mainly due to their high effectiveness and low toxicity.

The aim of the present study was to evaluate the safety and antiproliferative potential of Bulgarian rose essential oil in in vitro model of colorectal cancer (HT-29 and HCT-8 cell lines). In silico experiments were used to elucidate some of the mechanisms of activation of different pathways leading to apoptosis. In vitro and in silico studies can provide initial information about the possible biological effects of possible biological effects of BREO and the mechanisms by which they may be induced. The information obtained from these studies can help us determine the starting doses for future in vivo studies in experimental animals. In addition, the availability of preliminary information can contribute to a significant reduction in the number of experimental animals necessary for performing additional in vivo experiments to study the antitumor activity of BREO.

## 2. Materials and Methods

### 2.1. Collection and Storage of Plant Material

The flowers of the oil-bearing rose—Rosa damascena Mill were collected during flowering in May 2024 from the experimental field of the Institute of Roses and Aromatic Plants (IRAP), Agricultural Academy, Kazanlak, Bulgaria, in compliance with relevant national regulations. IRAP holds the intellectual property rights to the plant material cultivated within its experimental field. See “Official variety list of varieties if tobacco and vine, fruit medical and aromatic plant species accepted for certification and marketing on the territory of the Republic of Bulgaria 2025”; link: https://iasas.government.bg/att/OSL%202%20-%202025%20May%201.pdf (accessed on 12 May 2025). Rose flowers were collected during the rose picking season from 4 to 8 AM. in the IV–V developmental phase [29]. Fresh flowers were processed immediately using the microdistillation method in a Clevenger apparatus to extract rose oil.

### 2.2. Quantitative Determination of Rose Oil

The essential oil was obtained through water distillation in laboratory Clevinger type distillation equipment (Labbox, Barcelona, Spain) [30,31]. The distillation was carried out for 2 h at a module of 1:4 (ratio of rose flowers to water). The essential oil was measured in milliliters and was converted into weight percentages using the following formula: Y = (V × ρ × 100)/W, where Y is the yield of essential oil in %; V—the volume of the obtained essential oil (ml); ρ—the density of the rose oil (g/cm^3^) and W—the weight of the rose flower (g). The samples were stored in closed glass vials at 4 °C for subsequent analysis.

### 2.3. Gas Chromatographic (GC) Analysis

The gas chromatographic (GC) analysis of the BREO was performed on the Agilent 7820A Gas Chromatography with Flame Ionization Detector (GC FID) System. Standard calibration mixture of n-alkanes C_8_–C_27_ in hexane (≥98% purity—Honeywell, Riedel-de Haën TM) was used to enable quantification of major and minor components. The GC System is equipped with a non-polar capillary column EconoCapTM ECTM-5 (30 m × 0.32 mm × 0.25 µm film of 5% phenyl, 95% methylpolysiloxane). The split ratio was 1:10, the inlet temperature was 250 °C, and the FID temperature was set from 60 °C to 300 °C through a controlled program. Synthetic air (mixture of Nitrogen (N_2_) and Oxygen (O_2_)—80:20%) was used as a carrier gas and Hydrogen (99.999% purity) for the FID. Data was processed using Agilent chromatography data systems (CDS) software—Open Lab CDS—Agilent RapidControl.NET (version 9.2).

### 2.4. Cell Lines, Maintenance and Treatment

BALB 3T3 clone A31, ATCC: CCL-163™, murine embryonic fibroblast cells were used to perform the cytotoxicity and phototoxicity tests. These embryonic cells are highly sensitive to toxic substances. Therefore, they are most often used to study the safety of new chemicals and natural products. MCF-12F, ATCC: CRL-3599™ is non-tumorogenic epithelial cells. These cells are isolated from healthy tissue and have retained all mechanisms and molecular pathways for activating apoptosis, modulating the cell cycle, DNA repair systems, etc. Therefore, they are a suitable model of healthy tissue. HT-29 (ATCC: HTB-38™) cells are a suitable in vitro model for studying colorectal cancer, as they possess genetic and molecular characteristics (TP53 mutation, APC mutation and deregulated c-myc expression) typical of colorectal adenocarcinomas. HCT-8 (ATCC: CLL-244™) cells are used to study gastrointestinal cancers in vitro and have applications in the discovery of new antitumor drugs. In addition, HCT-8 are highly tumorigenic and can form tumors in vivo. The cell lines were purchased from American Type Cultures Collection (ATCC, Manassas, VA, USA). All cell cultures were maintained at 75 cm^2^ plastic dishes in Dulbecco’s Modified Eagle’s Medium (DMEM) high glucose (4.5 g/L glucose) supplemented with 10% fetal bovine serum (FBS), 1% sodium pyruvate, 100 U/mL penicillin and 0.1 mg/mL streptomycin at 37 °C in a humidified 5% CO_2_ atmosphere. Cells were maintained in the exponential growth phase after trypsinization and were plated at 1 × 10^4^ cells/well in 96-well cell culture microplates. After 24 h of cultivation, under the mentioned conditions, the cells were treated with the test substances, according to the specific experimental setup.

### 2.5. In Vitro Safety Test

The safety test was performed according to the European Union Reference Laboratory for Alternatives to Animal Testing, and accepted for regulatory use to detect the phototoxic potential of substances (Guidelines for the Testing of Chemicals, Section 4, Test No. 432 of the Organisation for Economic Co-operation and Development). Cell viability was measured using the neutral red uptake (NRU) assay [32]. The NRU assay measures cell viability based on the ability of viable cells to incorporate and bind the supravital dye neutral red in lysosomes. Then, a neutral red desorb solution is added, which dissolves the dye taken up by the lysosomes. The measured optical density of the solution is directly proportional to the cell viability. The cytotoxicity and phototoxicity tests were carried out on BALB 3T3 cells in 96-well microplates (1 × 10^4^ cells/well). After 24 h of incubation, the essential oil was added at increasing concentrations (from 4 to 1000 μg/mL). The phototoxicity test was performed in parallel on the second 96-well plate, which was irradiated (+Irr) with a dose of 2.4 J/cm^2^ using a solar light simulator Helios-iO (SERIC Ltd., Tokyo, Japan) for 10 min. After 24 h of incubation, culture medium containing Neutral Red was added. After 3 h of incubation, the wells were washed with a solution of PBS pH 7.4, and a solution of ethanol, acetic acid and distilled water in a ratio of 49:1:50 was added. The optical density (OD) was measured using a TECAN microplate reader at λ = 540 nm. Cytotoxicity is presented as % relative to the negative control (untreated cells).

### 2.6. In Vitro Antiproliferative Activity

The antiproliferative activity of the essential oil was determined in non-tumorogenic epithelial cells (MCF-12F) and tumor cell lines (colorectal cancer cell line HT-29 and gastrointestinal adenocarcinoma cells HCT-8) by MTT dye reduction assay [33]. The MTT assay is based on the enzymatic reduction of a yellow tetrazolium salt (MTT) by mitochondrial dehydrogenases in living cells, resulting in the formation of purple, insoluble formazan crystals. The amount of formazan formed is proportional to the metabolic activity of the cells, allowing for the quantitative determination of cell viability. Cells were plated at 1 × 10^3^ cells/100 µL/well of 96-well plates and incubated in a thermostat under standard conditions for 24 h. Cells were then incubated with the essential oil for 72 h. The optical density of the formazan was measured at λ = 540 nm using a microplate reader. Antiproliferative activity was expressed as % from the negative control (untreated cells). The IC_50_ values (50% inhibitory concentration) were determined by nonlinear regression analysis.

### 2.7. Clonogenic Assay

A clonogenic assay was performed to evaluate the effect of rose oil on colony formation by cancer cells. The tumor cells (500 cells/well) were cultured in a six-well tissue culture plate and incubated at 37 °C, 5% CO_2_ in humidified air for 24 h. Then, the rose oil was added at increasing concentrations (from 60 to 1000 µg/mL) into the respective wells and the plates were incubated for 10 days. The media was changed every alternate day and the plates were observed for colony formation under an inverted microscope. The colonies were stained with MTT solution before fixation. The percentage of colony formation inhibition was calculated for each concentration of rose oil relative to the negative control (untreated cells).

### 2.8. Statistical Analysis

The statistical analysis of the results was performed using one-way ANOVA followed by Bonferroni’s post hoc test by GraphPad Prism version 8 software (San Diego, CA, USA). *p* < 0.05 was accepted as the lowest level of statistical significance. All results are presented as mean ± SD.

### 2.9. Acridine Orange/Ethidium Bromide Live/Dead Staining

A fluorescence microscopy study was performed to further analyze the effects of the rose oil on cancer and non-cancerous cells. Cells were grown in 24-well plates. Essential oil was added to the cell cultures for 72 h at a concentration of 250 µg/mL. The cells were subjected to cytomorphological examination after staining with acridine orange (AO) and ethidium bromide (EB). Both fluorescent dyes were dissolved in PBS at a concentration of 5 µg/mL and applied to the cells as previously described [34].

### 2.10. Nuclear Morphology Analysis

The nuclear morphology of cells treated with rose oil was investigated by staining with the DNA-binding fluorescent dye 4′, 6 diamino-2phenylindole (DAPI, AppliChem, Darmstad, Germany). Cells were cultivated in 24-well plates and treated with essential oil (250 μg/mL) for 72 h. Following incubation, the cells were washed twice with PBS to remove detached cells. The cells were then fixed with ice-cold methanol. DAPI staining involves diluting the DAPI stock solution to a working concentration (300 nM in PBS), adding the DAPI working solution to the cells, and incubating for 5 min in the dark. This is followed by washing with PBS to remove excess dye. Cells were stored in the dark until examination with a fluorescence microscope (Leica DM 5000B, Wetzlar, Germany).

### 2.11. In Silico Molecular Docking Simulations

The 3D crystal structure of MAPKP38α (3ZS5) and dimer complex of the human glucocorticoid receptor ligand-binding domain (1M2Z) were retrieved from the Protein Data Bank (https://www.rcsb.org/structure/, accessed on 20 July 2025). The 3D structure formulas of geraniol (CID: 637566), citronellol (CID: 8842), nerol (CID 643820) and linalool (CID: 6549) were downloaded from NCBI PubChem (https://pubchem.ncbi.nlm.nih.gov/, accessed on 12 July 2025). The ligands and receptors were prepared before docking using the BIOVIA Discovery Studio Visualizer 2025 (v25.1.0.0) and AutoDock Vina (v4.2.6) [35,36], MGLTool (v1.5.7), including removing water, adding hydrogen, assigning charges and generating molecular surfaces. Docking simulations were performed using the AutoDock Vina [37]. The docking pose with the lowest binding energy (kcal/mol) was considered the most suitable. BIOVIA Discovery Studio Visualizer software was used to visualize the ligand and receptor interaction in 3D and 2D structures, respectively.

## 3. Results

The BREO has been studied quantitatively and qualitatively. The BREO was obtained by microdistillation in a Clevenger apparatus. A total of 300 g of fresh flowers were used for each distillation. The calculated average yield of essential oil in percent was obtained from six independent distillations (Table 1). A good result for the yield of the rose oil is considered 0.06 mL. Our results showed the average volume of the essential oil obtained was 0.107 ± 0.008 mL and the average yield was 0.046 ± 0.004%.

### 3.1. Gas Chromatographic Analysis of BREO

The gas chromatographic analysis of the BREO was performed on an “Agilent 7820A GC FID” gas chromatographic system. In accordance with the ISO norm 9842:2024, a FID was selected, which provides a broad detection range and robustness [38]. Quantification of the main compounds was achieved by measuring peak area, without applying any correction factor. Data was processed using Agilent chromatography data systems (CDS)—Open Lab CDS—Agilent RapidControl.NET (Figure 1). The chromatographic profile of Rosa damascena Mill essential oil revealed a number of peaks that represented the various volatile components included in the essential oil.

Gas chromatographic analysis enabled the identification of 20 constituents, and is in conformity with the ISO 9842:2024 standard [38,39]. Table 2 shows the retention time (min) and peak height as a relative percentage of detected compounds. The main components identified were terpene alcohols: geraniol, 28.73 % (reference values (RVs): 14.0–22.0%); citronellol, 21.5 % (RVs: −20.0–34.0%); and nerol, 5.51% (RVs: −5.0–12.0%).

### 3.2. Safety Test

An in vitro BALB 3T3 NRU assay was used to determine the cytotoxicity/phototoxicity of the BREO. In this test, the phototoxic drug chlorpromazine was used as a positive control. The resulting dose–response curves describing cyto- and phototoxicity are presented in Figure 2. The CC_50_ (50 % cytotoxic concentration) values were calculated and are presented in Table 3. Photo irritation factor (PIF) was used to assess the phototoxic potential of the rose essential oil. The PIF factor represents the ratio between the CC_50_ values of dark (non-irradiated or −Irr) and irradiated (+Irr) microplates, PIF = (CC_50_ – Irr)/(CC_50_ + Irr). The PIF factor for essential oil is lower (0.92), indicating a high level of photo safety.

### 3.3. Antiproliferative Activity

The BREO was studied for antiproliferative activity with an MTT-dye reduction assay. Cell cultures were incubated with the BREO in twofold increasing concentrations (from 7.5 to 1000 μg/mL) for 72 h. The results are presented graphically in Figure 3. The results show that the antiproliferative effect of BREO on the studied cell lines is of a dose–response type. The 50% inhibitory concentration of cell proliferation (IC_50_) values and selectivity index (SI) were presented in Table 4. At 72 h of incubation, BREO most strongly inhibited the proliferation of the tumor cell line HT-29 (IC_50_ = 290.45 ± 10.79 µg/mL). Significantly lower antiproliferative activity was observed in the normal cells MCF-12F (IC_50_ = 383.90 ± 34.75 µg/mL) and tumor cells HCT-8 (IC_50_ = 363.67 ± 12.43 µg/mL). The selectivity of BREO towards tumor cell lines is significantly lower than the positive control (Cisplatin).

### 3.4. Clonogenic Assay

We determined the efficacy of BREO to inhibit colony formation of the tested cell lines by a clonogenic assay. The cells were incubated in the presence of BREO at different concentrations (from 60 to 1000 µg/mL) for 10 days. The cells were then stained with MTT (Figure 4A,C). The percentage of colony formation inhibition was calculated for each concentration of BREO relative to the negative control (untreated cells) (Figure 4B,D). The calculated concentration at which we observed 50% inhibition of colony formation was 163.79 ± 10.25 µg/mL for HT-29 cells. This concentration is 43% lower than the IC_50_ value determined for the antiproliferative activity of BREO in HT-29 cells. In the HCT-8 cell line, significantly lower activity of BREO was observed (IC_50_ = 298.62 ± 6.14 µg/mL). This is 18% less than the IC_50_ value determined in the antiproliferative activity test.

### 3.5. Microscopic Analysis

The cellular and nuclear morphologies of BREO-treated cells were examined by fluorescence microscopy (Figure 5 and Figure 6). The cells were stained with acridine orange/ethidium bromide to assess the ability of the BREO to induce cell death (Figure 5).

The staining with AO/EB allows discrimination between the dead and viable cells, based on the differences in their permeability for the fluorescent dyes. Viable cells with intact cytoplasmic membranes are not permeable for the EB, while AO passes through the membranes and emits green fluorescence. Late apoptotic and dead cells are stained in orange and red due to the increase of membrane permeability that enables the cell internalisation of EB. In all tested cell lines, the control untreated cells showed green staining and numerous mitotic cells were observed (Figure 5a–c). The treatment of HT-29 and HCT-8 cells with BREO induced marked morphological alterations (Figure 5e,f). The cell monolayer confluency was greatly reduced and individual orange-red-stained dead cells were noticed. In non-cancerous MCF-12F cell cultures, a slight ruffling of the monolayer was found; however, dead cells were not detected (Figure 5d). These results are consistent with the data from the MTT assay and further demonstrate the selectivity of the rose oil effect towards the cancer cells.

The nuclear morphology alterations induced by the rose oil treatment were examined by DAPI staining (Figure 6).

The nuclei of the control untreated cells were with morphological features typical for the respective cell lines and showed homogeneous blue staining (Figure 6a–c). In MCF-12F cell cultures treated with BREO, nuclear morphology was similar to that of the control and cells in a phase of mitosis were frequently observed (Figure 6a,d). In contrast, in the BREO-treated cancer cell cultures, mitotic cells were noticed very rarely. Moreover, nuclei with typical apoptotic morphology, such as chromatin condensation, nuclear fragmentation and formation of apoptotic bodies were detected (Figure 6e,f). The results of fluorescent microscopy analysis demonstrate that the detected antiproliferative activity of the BREO is due to both inhibition of cell growth and induction of apoptotic cell death.

### 3.6. In Silico Molecular Docking Simulations

Analysis of each binding pose was conducted for binding score and presence of molecular interactions between geraniol, citronellol and nerol targeting MAPKP38α- and glucocorticoid receptor ligand binding domain (GR-LBD) receptors, respectively. The ligands (i.e., geraniol, citronellol and nerol) were subjected to molecular docking with all target regions of the gD molecule using AutoDock Vina (v. 4.2.6). The dock preparation function of MGLTools (v. 1.5.7) was utilized to prepare each protein for docking that included hydrogens and Gasteiger charges. Forcefield calculations were applied to analyze the ligand for hydrogen atoms, applicable charges and rotatable bonds. The grid area for MAPKP38α interactions and GR-LBD with REO ingredients geraniol, citronellol and nerol was defined for the targeting ligands. The simulation of docking was conducted with Genetic Algorithm by binding energy minimization and evaluation with RMSD analysis using the ligand atoms only. The number of conformations was fixed to 10. The top-ranked poses of each ligand with the lowest binding energies were found to be within the definite binding sites of both receptors (MAPKP38α and GR-LBD). We have chosen the lowest energy of binding complexes from 10 conformation possibilities of docking with MAPKP38α and GR-LBD for each ligand. The main interactions were hydrophobic and through hydrogen bonding. The studied compounds of REO, which are terpene alcohols, indicate that only the hydroxyl groups are indispensable for inhibiting the activity of MAPKP38α and GR-LBD in cancer cells. The lowest binding energies of geraniol, citronellol and nerol with the interaction interface of MAPKP38α and GR-LBD receptor molecules are shown in Table 5.

The interactions of geraniol, targeting the MAPKP38α and GR-LBD, are presented in Figure 7 and Figure 8, and citronellol and nerol in Figure 9 and Figure 10, respectively. Figure 7 shows MAPKP38α- and GR-LBD complexes with geraniol in the most stable conformation—a common view. In Figure 8A close view of the interaction interface is shown in 3D and 2D images. Figure 9 and Figure 10 exhibit a close view of the interaction interface of MAPKP38α- and GR-LBD complexes with citronellol and nerol, respectively.

The docking study with geraniol exhibited –7.07 and –6.34 kcal/mol, which is considered a good, potentially strong binding affinity towards MAPKP-38α and GR-LBD, respectively (Table 1). The lowest binding affinity of geraniol with MAPKP38α than GR-LBD was observed in comparison with the other two ligands, citronellol and nerol. Citronellol and nerol showed appreciable binding affinities towards MAPKP-38α and GR-LBD.

## 4. Discussion

The yield and quality of essential oils and other biologically active substances of plant origin depend largely on the soil and climatic characteristics of the geographical region from which they are extracted [40,41]. In the context of accelerated climate change, the rainfall amounts and average daily temperatures are changing, which significantly affect the yield and quality of rose oil. That is why it is important to conduct annual qualitative and quantitative tests of the extracted rose oil in the relevant year. Our results showed that the average yield of BREO for 2024 was 0.046 ± 0.004%. In comparison with a previous study conducted in 2010, the BREO contents for rose cultivars developed from the Institute of Roses and Aromatic Plants were similar (cv. Svezhen—0.054%; cv. Iskra—0.047%; cv. Janina—0.045% and cv. Elejna—0.052%) [42]. The gas chromatographic analysis of rose oil isolated in 2024 showed that the concentrations of the main chemical components were within the limits specified by the ISO 9842:2024 standard [38]. Only the concentration of geraniol was higher by 6.7% compared to the upper limit specified in the standard. Compared to a previous study conducted in 2009, the difference in the concentration of geraniol was more than 9% [43]. In contrast, the concentration of citronellol measured in both studies varied within very small limits (<3%). Due to the large amount of citronellol and geraniol in BREO, these two substances determine its biological activity. They have been shown to affect peroxisome proliferator-activated receptor (PPAR), cyclooxygenase-2 (COX-2) and suppress LPS-induced COX-2 expression in cell culture assays [44]. Minor components include rose oxide, eugenol and other esters and alcohols. Hydrocarbons were represented by saturated aliphatic homologs with an odd number of carbon atoms (nonadecane, heneicosane, heptadecane and tricosan). Linalool is present in relatively low concentrations in rose oil (from 0.5% to 2.75%) but has significant biological activity. Linalool has been found to dose-dependently inhibit cell growth in a number of in vitro and in vivo tumor models: hepatocellular carcinoma, lung carcinoma, prostate cancer, leukemia, cervical cancer, breast cancer, etc. [17,45,46,47,48,49]. It has been shown that the antiproliferative concentrations of linalool strongly depend on the cell line and on whether it is administered alone or in combination with other conventional drugs.

Studies by other scientific teams from Bulgaria [50], Iran [51] and Turkey [52] show different aspects of the safety tests of rose oil. Cytotoxicity and genotoxicity tests were performed on other cell lines (Beas-2B, NIH3T3, lymphocytes, etc.). The results obtained vary widely depending on the geographical origin of the rose oil and the methods used for safety assessment. The comparative analysis showed that BREO has the highest level of safety compared to rose oil from Iran and Turkey. Furthermore, the photosafety test showed the absence of phototoxicity in BREO. This is an important result since we did not find any studies in the literature related to the photosafety of BREO. The low cytotoxicity and high level of photosafety allow BREO to be used in the field of cosmetics and pharmacy without the risk of side effects.

The antiproliferative effect in the three studied cell lines was observed at two times lower concentrations than the CC_50_ determined in mouse embryonic fibroblasts. The low selectivity indicates that the BREO also affects normal cells (MCF-12F), inhibiting their growth. It is likely that the substances with the highest concentration in rose oil (geraniol, 28.73% and citronellol, 21.5%) inhibit cell proliferation. There is evidence that geraniol can inhibit cell proliferation by inducing S-phase cell cycle arrest in Caco-2 cells (colorectal adenocarcinoma) [53]. In addition, geraniol can induce apoptosis in tumor cells [54] and affect various signaling pathways regulating cell proliferation [55]. Citronellol inhibits cell proliferation by inducing cell cycle arrest in different phases (G1 and G2/M) in different tumor cell lines [56]. Citronellol can induce mitochondrial-mediated apoptosis through activation of proapoptotic factors in tumor cells [57]. The results of the clonogenic assay showed a higher potential of BREO to inhibit cell proliferation and colony formation in colorectal carcinoma (HT-29) when treatment is performed for a longer period of time. In other scientific studies, increased sensitivity of the HT-29 cell line to 5-fluorouracil has been observed with long-term treatment [58]. It is suggested that this increased sensitivity of HT-29 cells to 5-fluorouracil is associated with DNA-directed toxicity [59]. The results of fluorescent microscopy analysis demonstrate that the detected antiproliferative activity of the BREO in colorectal carcinoma is due to both inhibition of cell growth and induction of apoptotic cell death.

For our in silico experiment, we chose the ATP pocket of the MAPK p38-α binding interface and the GR-LBD (547–735 amino acid residues) to target the ligands (geraniol, citronellol and nerol). Based on consensus binding affinities, critical interactions and binding, geraniol, citronellol and nerol might act as a possible binding partner of MAPKP38α, which can modulate their functional activity via decreasing the accessibility of substrate. Geraniol may have a high potential to inhibit their function, thereby acting as a putative therapeutic agent for cancer. The important strategy for the design of novel kinase inhibitors is selectivity, through maximization of the number of interactions throughout the ATP pocket and the exploitation of specific features in the active site.

The discovery that inhibition of p38 activity has potential for the anti-inflammatory treatment of COPD is likely to trigger renewed interest in modulating the activity of the p38 signaling pathway using small-molecule inhibitors. The conformational flexibility of the p38 protein was combined with protein kinase selectivity profiles and contact-residue information to generate an in-depth understanding of the relationships between compound chemical structure, compound-induced protein conformation and selectivity [60]. On the other hand, oxidative stress is one of the causes of carcinogenesis and consequently, the antioxidant activity strategy for searching bioactive compounds with such capacity. A number of articles have been reported for the radical-scavenging activities and radioprotective properties of BREO [61,62]. The importance of the balance between anti-oxidant and pro-oxidant activity limits the potential of the strategies for anti-tumor therapeutics. Inflammation is initiated by oxidative stress, which induces cytokine production in response to an external or pathophysiological agent [63]. Cytokines and ROS may activate different lymphocytes to encounter inflammation [64,65]. In classical GR signaling, GR can interact with other transcription factors such as AP-1 and NF-κB, which are involved in inflammation. This is important because MAPK p38α is part of inflammatory signaling [66].

The GR is a modular protein containing an N-terminal transactivation domain (NTD), a central DNA-binding domain (DBD), a C-terminal ligand-binding domain (LBD) and a flexible ‘hinge region’ separating the DBD and the LBD. The NTD has a strong transcriptional activation function (AF1), which allows for the recruitment of coregulators and transcription machinery. In addition to its role in ligand-dependent activation function (AF-2) that is tightly regulated by hormone binding [67]. A direct mechanism of action involves homodimerization of the receptor, translocation via active transport into the nucleus, and binding to specific DNA elements. The two GR LBD monomers in each asymmetric unit are arranged in a dimer configuration. The two GR monomers show a C2 symmetric packing arrangement in which either LBD can be superimposed on the other by rotating 180 degrees around the 2-fold axis. The central hydrophobic interface is made up of reciprocal interactions between residues P625 and I628 in the b turn of strands 3 and 4. Surrounding this core hydrophobic interface is an extensive network of hydrogen bonds mediated by the extended strand between helices 1 and 3 (residues 547–551) and the last residue of helix 5 (Q615). In particular, residues 547–551 from each LBD, resembling two anti-parallel b strands, are in excellent geometry to form four hydrogen bonds. These hydrogen bonds may also play a key role in stabilizing the GR dimer configuration.

The results of our studies showed a high level of safety and antiproliferative activity of BREO in colorectal carcinoma. These are indicators of the potential of BREO to be used in colon cancer in combination with standard chemotherapeutics or as an additional therapeutic agent. The results of our in vitro experiments provide information about the biological properties of the BREO, but it is insufficient. The results of our in vitro experiments provide information about the biological properties of the BREO, but it is insufficient. The presented study also has some limitations. Each batch in different years shows partial variations in the composition and amount of active components. This leads to certain changes in biological activities. This is due to the change in climatic conditions during the year (amount of rain, temperature, light, etc.), which leads to changes in the synthesis of components in plant cells. Important for a good yield is also how and under what conditions the period of collection and storage of rose petals occurs. In terms of biological activity, additional in vitro experiments are needed to study the mechanism of antiproliferative activity. Flow cytometric analysis of the cell cycle and apoptosis would provide important additional information. Another limitation of our study is that it was conducted only in vitro and in silico conditions. In order to confirm the antitumor activity against colorectal carcinoma, additional in vivo studies on experimental animals are needed.

## 5. Conclusions

The studied BREO meets the requirements of the ISO 9842:2024 standard in terms of chemical composition. The safety test showed low cytotoxicity and a high level of photosafety. The tests for antiproliferative activity and inhibition of colony formation showed that BREO exhibits its activity in a dose- and time-dependent manner. Fluorescence microscopy showed reduced confluence of the monolayer and the presence of dead cells in the treated tumor cell cultures. In addition, reduced mitotic activity and the presence of cell nuclei with typical apoptotic morphology were observed in the treated tumor cells. Analysis of the results showed that the in vitro model of colorectal carcinoma HT-19 is highly sensitive to BREO. In silico experiments confirmed the potential of BREO to induce an antiproliferative effect and activate various apoptosis pathways. The data indicate the possible stages at which the components of rose oil may act to stop the progression of cancer, and thus highlight the potential of BREO as a candidate for additional therapy of colorectal adenocarcinoma.

Additional experiments with a suitable animal model are needed to further investigate the anticancer activity of BREO. We plan to continue our research using an in vivo mouse model of colorectal carcinoma to investigate the antitumor activity of BREO. The survival time of tumor-bearing animals with oral administration of BREO will be recorded. In addition, histological, immunological and biochemical analyses of biological samples taken from the experimental animals will be performed in order to study in more detail the effect of BREO on healthy and tumor-implanted animals.

## Figures and Tables

**Figure 1 cimb-47-00649-f001:**
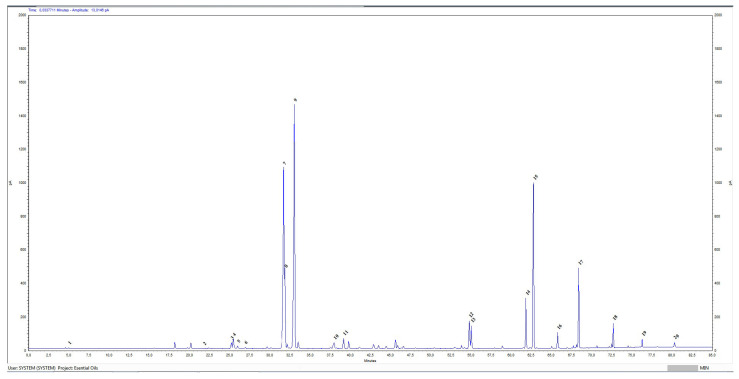
Chromatographic profile of Rosa damascena Mill essential oil. X-axis—retention time (RT)—minutes; y-axis—picoampere (pA). Sequence number by retention time of identified compounds: 1—Ethanol; 2—Limonene; 3—Linalool; 4—Phenylethanol; 5—Cis-rose oxide; 6—Trans- rose oxide; 7—Citronellol; 8—Nerol; 9—Geraniol; 10—Eugenol; 11—Methyl eugenol; 12—Heptadecane; 13—Farnesol; 14—Nonadecene; 15—Nonadecane; 16—Eicosane; 17—Heneicosane; 18—Tricosane; 19—Pentacosane; 20—Heptacosane.

**Figure 2 cimb-47-00649-f002:**
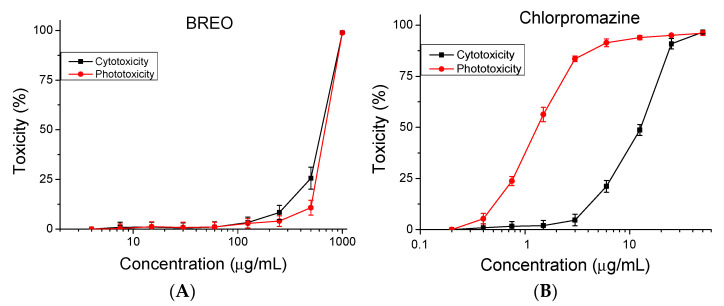
Cytotoxicity/phototoxicity of (**А**) BREO and (**B**) chlorpromazine (positive control) determined in BALB 3T3 cells. Compounds were tested in two-fold increasing concentrations, *n* = 6.

**Figure 3 cimb-47-00649-f003:**
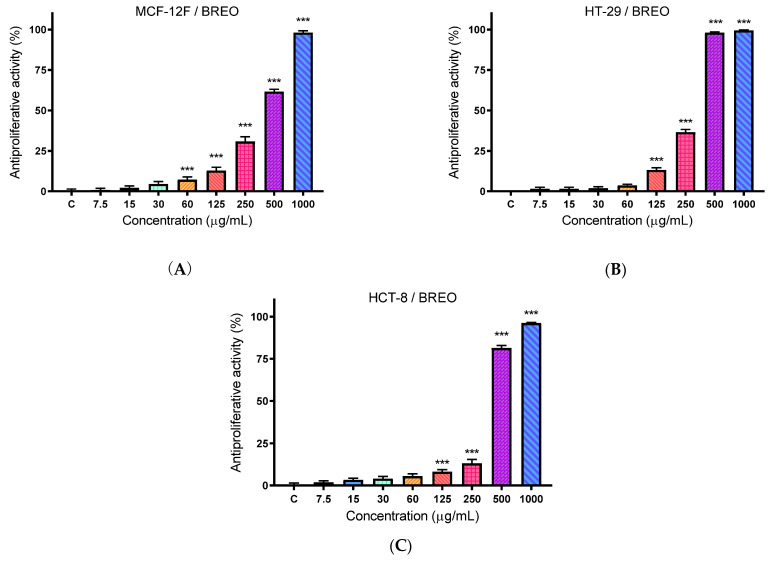
Antiproliferative activity of BREO in the: (**A**) MCF-12F cells (in vitro model of healthy tissue), (**B**) HT-29 cells and (**C**) HCT-8 cells, *** *p* < 0.001 (one-way ANOVA test), *n* = 6.

**Figure 4 cimb-47-00649-f004:**
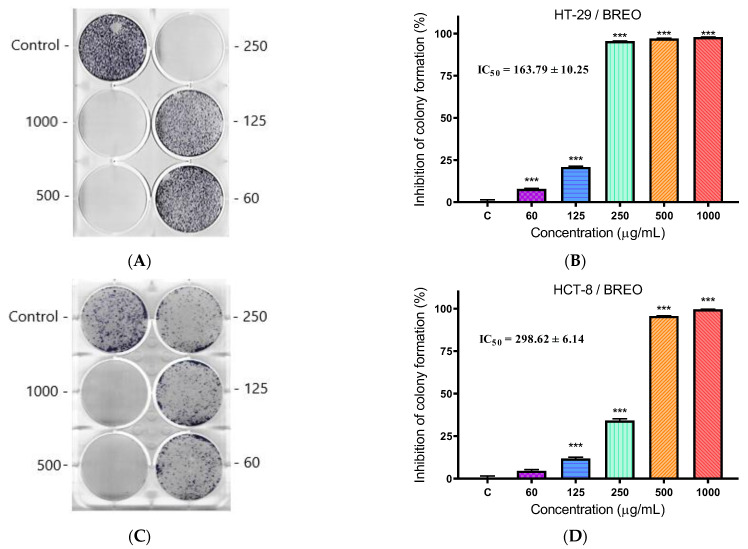
Inhibition of colony formation of BREO in the HT-29 and HCT-8 cells. (**A**,**C**) Coloring with MTT dye. (**B**,**D**) Graphical representation of the results in % inhibition of colony formation compared to the negative control (untreated cells), *** *p* < 0.001 (one-way ANOVA test).

**Figure 5 cimb-47-00649-f005:**
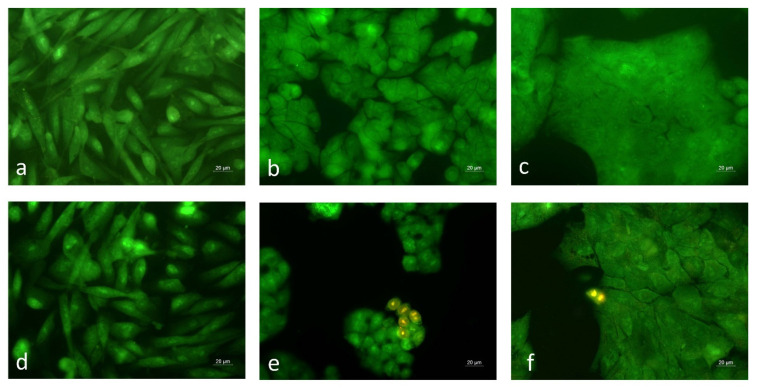
Cytomorphological alterations in human cancer and non-cancer cell lines treated for 72 h with BREO. (**a**–**c**) control untreated cells; (**d**–**f**) cells treated with BREO (250 μg/mL); (**a**,**d**) MCF-12F cells; (**b**,**e**) HT-29 cells; (**c**,**f**) HCT-8 cells; acridine orange/ethidium bromide staining; fluorescent microscopy; objective 40×.

**Figure 6 cimb-47-00649-f006:**
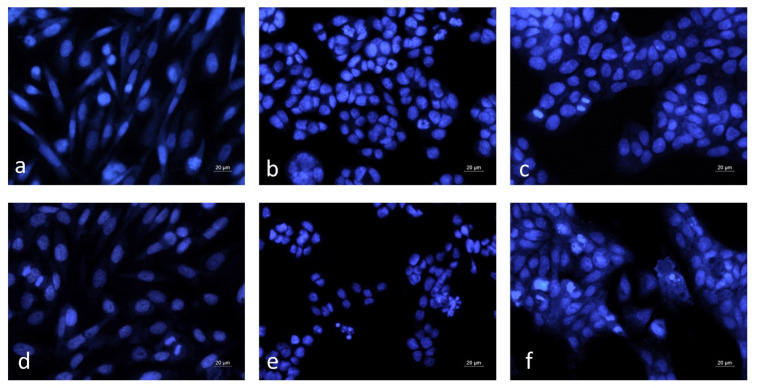
Nuclear morphology alterations in human cancer and non-cancer cells induced by 72 h treatment with BREO. (**a**–**c**) control untreated cells; (**d**–**f**) cells treated with BREO (250 μg/mL); (**a**,**d**) MCF-12F cells; (**b**,**e**) HT-29 cells; (**c**,**f**) HCT-8 cells; DAPI staining; fluorescence microscopy; objective 40×.

**Figure 7 cimb-47-00649-f007:**
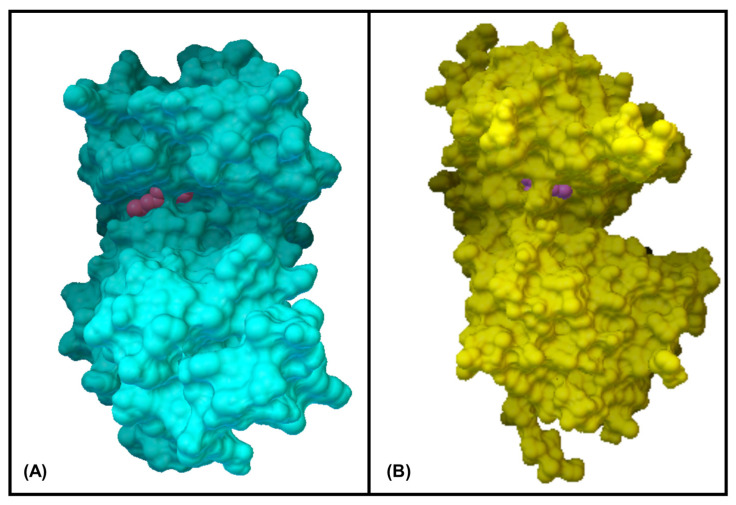
MAPKP38α- and GR-LBD complexes with geraniol in the most stable conformation. (**A**)—3D structure of MAPKP38α and geraniol complex, (**B**)—3D structure of GR-LBD and geraniol complex.

**Figure 8 cimb-47-00649-f008:**
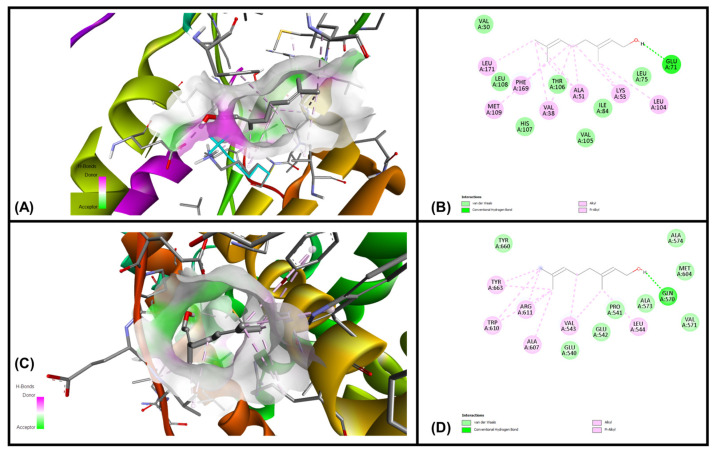
The interaction interface of MAPKP38α and GR-LBD with geraniol. (**A**) Three-dimensional structure of the interaction interface of MAPKP38α and geraniol. (**B**) Two-dimensional structure of the interaction interface of MAPKP38α and geraniol. (**C**) Three-dimensional structure of the interaction interface of GR-LBD and geraniol. (**D**) Two-dimensional structure of the interaction interface of GR-LBD and geraniol.

**Figure 9 cimb-47-00649-f009:**
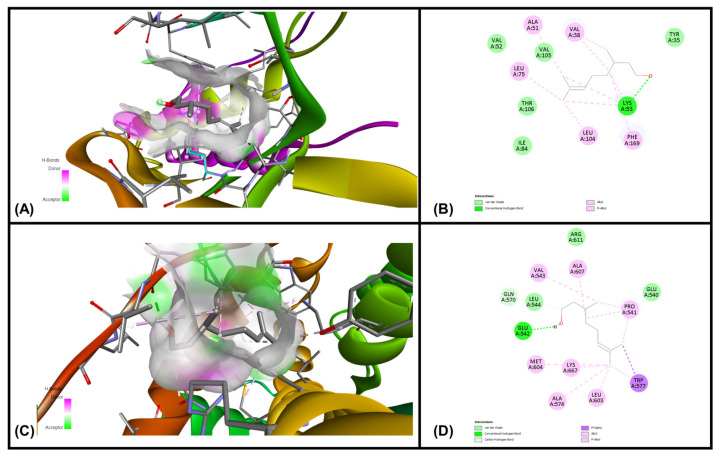
The interaction interface of MAPKP38α and GR-LBD with citronellol. (**A**) Three-dimensional structure of the interaction interface of MAPKP38α and citronellol. (**B**) Two-dimensional structure of the interaction interface of MAPKP38α and citronellol. (**C**) Three-dimensional structure of the interaction interface of GR-LBD and citronellol. (**D**) Two-dimensional structure of the interaction interface of GR-LBD and citronellol.

**Figure 10 cimb-47-00649-f010:**
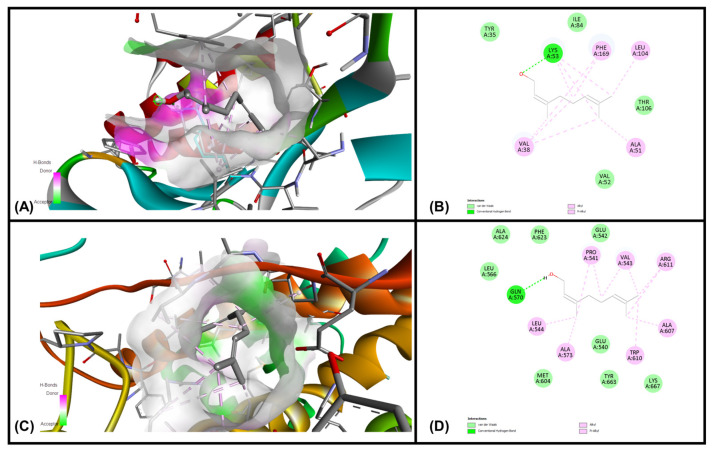
The interaction interface of MAPKP38α and GR-LBD with nerol. (**A**) Three-dimensional structure of the interaction interface of MAPKP38α and nerol. (**B**) Two-dimensional structure of the interaction interface of MAPKP38α and nerol. (**C**) Three-dimensional structure of the interaction interface of GR-LBD and nerol. (**D**) Two-dimensional structure of the interaction interface of GR-LBD and nerol.

**Table 1 cimb-47-00649-t001:** Quantitatively evaluation of BREO harvested in May of 2024.

Sample	I	II	III	IV	V	VI
Ratio	1:4	1:4	1:4	1:4	1:4	1:4
Volume (mL)	0.11	0.10	0.10	0.12	0.10	0.11
Yield (%)	0.0475	0.043	0.043	0.0518	0.043	0.0475

**Table 2 cimb-47-00649-t002:** Rose oil composition.

№	Components	Molecular Formula	Retention Time (RT)	Relative %	Reference Values: ISO 9842:2024
1	Ethanol	C_2_H_6_O	4.996	0.06	≤3.0%
2	Limonene	C_10_H_16_	21.796	0.05	
3	Linalool	C_10_H_18_O	25.216	0.53	
4	Phenylethanol	C_8_H_10_O	25.462	0.96	≤2.5%
5	Cis-rose oxide	C_10_H_18_O	25.987	0.23	
6	Trans- rose oxide	C_10_H_18_O	26.935	0.13	
7	Citronellol	C_10_H_20_O	31.726	21.50	20.0–34.0%
8	Nerol	C_10_H_18_O	31.871	5.51	5.0–12.0%
9	Geraniol	C_10_H_18_O	33.064	28.73	14.0–22.0%
10	Eugenol	C_10_H_12_O	37.987	0.85	
11	Methyl eugenol	C_11_H_14_O_2_	39.811	0.70	0.8–3.0%
12	Heptadecane	C_17_H_36_	54.805	2.29	1.0–2.5%
13	Farnesol	C_15_H_26_O	55.049	2.11	
14	Nonadecene	C_19_H_38_	61.846	3.93	1.5–4.0%
15	Nonadecane	C_19_H_40_	62.787	13.13	8.0–15.0%
16	Eicosane	C_20_H_42_	65.789	1.01	
17	Heneicosane	C_21_H_44_	68.410	4.87	3.0–5.5%
18	Tricosane	C_23_H_48_	72.698	1.29	
19	Pentacosane	C_25_H_25_	76.302	0.46	
20	Heptacosane	C_27_H_56_	80.324	0.39	

**Table 3 cimb-47-00649-t003:** Cytotoxicity of BREO in BALB 3T3 cells, values of CC_50_ and photo irritation factor.

Compounds	Mean CC_50_ ± SD (µg/mL)		PIF *
	−Irr	+Irr **	
BREO	629.72 ± 22.38	682.99 ± 14.39	0.92
Chlorpromazine ***	12.74 ± 0.82	1.31 ± 0.07	9.73

* PIF—photo irritation factor: PIF < 2 = not phototoxic, 2 < PIF < 5 = possible phototoxicity, PIF > 5 phototoxic; ** Irr—irradiation, *** chlorpromazine (positive control).

**Table 4 cimb-47-00649-t004:** Average IC_50_ values and selectivity index (SI).

Compounds	Mean IC_50_ ± SD (µg/mL)			SI *	
	MCF-12F	HT-29	HCT-8	HT-29	HCT-8
BREO	383.90 ± 34.75	290.45 ± 10.79	363.67 ± 12.43	1.32	1.06
Cisplatin **	22.8 ± 1.5	8.2 ± 0.4	13.2 ± 1.2	2.78	1.73

* Selectivity index (SI), SI = IC_50_ (normal cells)/IC_50_ (tumor cells). ** Cisplatin—positive control.

**Table 5 cimb-47-00649-t005:** The main BREO compounds (geraniol, citronellol and nerol) binding energies for the MAPKP38α and glucocorticoid receptor ligand binding domain (GR-LBD).

Ligands	MAPKP38α (Kcal/mol)	GR-LBD (Kcal/mol)
Geraniol	−7.07	−6.34
Citronellol	−6.01	−5.28
Nerol	−5.75	−5.58

## Data Availability

The original contributions presented in this study are included in the article. Further inquiries can be directed to the corresponding author(s).

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
