# Peer review of "Evaluation of the Safety and Antiproliferative Activity of Bulgarian Rose Essential Oil: An In Vitro and In Silico Model of Colorectal Adenocarcinoma"

_cimb, 2025, doi:10.3390/cimb47080649_

Round 1
Reviewer 1 Report
Comments and Suggestions for Authors
This article focuses on the ability of Bulgarian rose oil to stop colorectal cancer cells from growing in vitro using HT-29 and HCT-8 cell lines. The methods applied were seems to be appropriate and fully described in the article. Similarly, the results sufficiently support the writing and the findings, and the reference list is also very significant. However, some minor important issues arise from the manuscript at its present state:
- Despite the fact that the study is adequate, I think that giving the results of other molecules studied theoretically will increase the quality of the article.
- The article should have a computational analysis to confirm the results obtained.
- I believe that the discussion could be expanded upon and the introduction should be revised to indicate the importance of advanced studies for this research.
Author Response
Response to Reviewer Comments
We would like to sincerely thank the reviewer for the valuable and constructive feedback, which has helped us improve the quality and clarity of our manuscript. We have carefully addressed each point, and the requested revisions have been incorporated into the revised version of the manuscript as detailed below.
Comment 1: Despite the fact that the study is adequate, I think that giving the results of other molecules studied theoretically will increase the quality of the article.
The article should have a computational analysis to confirm the results obtained.
Response 1:We agree with the reviewer that computational analysis and theoretical modeling would strengthen the findings and provide further insight into the mechanisms of action of the bioactive compounds in Bulgarian rose oil. In response to this reviewer comment, we included in our revised manuscript additional data from molecular docking analyses. The conducted analysis presents the interaction between the cellular MAPKP38α and GR-LBD and the three most active components of the composition of rose oil: geraniol, nerol and citronellol.
Comment 2: I believe that the discussion could be expanded upon and the introduction should be revised to indicate the importance of advanced studies for this research.
Response 2: The introduction has been revised and the discussion has been expanded.
Reviewer 2 Report
Comments and Suggestions for Authors
The manuscript is well written just a few points I have to make:
1- consider using one term for one specific item; rose oil , oil , essential oil; are probably referring to the same thing and might be confusing.
2- please explain what neutral red uptake assay measures and familiarize the readers
3- please give a brief description of the cell lines used for better understanding
4-please explain the process of MTT dye reduction briefly
5-in the colony staining please clarify whether the MTT staining was done after fixation or directly in culture
6- include information on DAPI staining
7- please specify the type of microscope for fluorescent microscopy
8- Please ensure the results is clearly tied to the discussion following because you have the results and discussion in one segment.
Author Response
Response to Reviewer Comments
We would like to sincerely thank the reviewer for the valuable and constructive feedback, which has helped us improve the quality and clarity of our manuscript. We have carefully addressed each point, and the requested revisions have been incorporated into the revised version of the manuscript as detailed below.
Comment 1: Consider using one term for one specific item; rose oil , oil , essential oil; are probably referring to the same thing and might be confusing.
Response 1: We agree and have revised the manuscript to consistently use the term "Bulgarian rose essential oil" (BREO) throughout the text to avoid confusion.
Comment 2: please explain what neutral red uptake assay measures and familiarize the readers
Response 2: A brief explanation has been added in the Materials and Methods section. The NRU assay measures cell viability based on the ability of viable cells to incorporate and bind the supravital dye neutral red in lysosomes. Then, a neutral red desorb solution is added, which dissolves the dye taken up by the lysosomes. The measured optical density of the solution is directly proportional to the cell viability.
Comment 3: please give a brief description of the cell lines used for better understanding
Response 3: We have included a brief description of cell lines in the Materials and Methods section to clarify their relevance as colorectal cancer models.
Comment 4: please explain the process of MTT dye reduction briefly
Response 4: MTT assay is based on the enzymatic reduction of the yellow tetrazolium salt (MTT) by mitochondrial dehydrogenases in metabolically active cells, leading to the formation of purple, insoluble formazan crystals. The amount of formazan formed is proportional to the metabolic activity of the cells, allowing for quantification of cell viability. A brief description of the test method principle has been added in the Materials and Methods section.
Comment 5: in the colony staining please clarify whether the MTT staining was done after fixation or directly in culture
Response 5: We have specified in the Materials and Methods section that MTT staining was performed before fixation of the colonies.
Comment 6: include information on DAPI staining
Response 6: A brief description of the DAPI staining procedure has been added to the Materials and Methods section.
Comment 7: please specify the type of microscope for fluorescent microscopy
Response 7: The model of the fluorescence microscope (Leica DM 5000B, Wetzlar, Germany) have been specified in the Materials and Methods section of the revised manuscript.
Comment 8: Please ensure the results is clearly tied to the discussion following because you have the results and discussion in one segment.
Response 8: We have separated the Results and Discussion into distinct sections for improved clarity and structure, as recommended.
Reviewer 3 Report
Comments and Suggestions for Authors
- Define abbreviations for the first time they appear in the text (e.g., GC FID, MTT, NRU…etc.).
- Line 26-31: Please specify the level of concentration.
- Line 31-32: A clear path for further research should be suggested in the conclusion.
- For clarity, the introduction's first lengthy paragraph should be broken up into smaller ones. Line 64 might be shown as the beginning of a new paragraph.
- Line 64-78: Numerous statements are provided without references.
- The molecular biology of colorectal cancer should be clearly reviewed in the introduction.
- The second paragraph in introduction is unclear and should be expand for clarity. For example, polyphenols in Line 80 and phytochemicals in Line 82 & 85 should be defined. Define chemotherapeutic drugs in Line 83-84 & 86.
- If the introduction discussed the molecular role of essential oils derived from different medicinal plants and the range of pathways impacted in colorectal cancer lines, that would be interesting.
- I would recommend adding a concluding paragraph to the introduction to explain the purpose of the paper in relation to what is currently understood by in vivo/vitro studies, thereby emphasizing the study's added value.
- Each of the results and the discussion should be presented in its own section.
- The discussion provides insufficient interpretation of the results. The section should be prepared by organizing information according to: the main findings and comparison of these findings with those reported in the literature, the strengths and weaknesses of the study and in relation to other studies, information about the present analyses and the implications of this study and future research directions.
- Please update as many of the references were outdated (Ref # 15-19).
Modest editing for language is required throughout the paper by a native English speaker.
Author Response
Response to Reviewer Comments
We would like to sincerely thank the reviewer for the valuable and constructive feedback, which has helped us improve the quality and clarity of our manuscript. We have carefully addressed each point, and the requested revisions have been incorporated into the revised version of the manuscript as detailed below.
Comment 1: Define abbreviations for the first time they appear in the text (e.g., GC FID, MTT, NRU…etc.).
Response 1: In the reviesed manuscript all abbreviations have been defined upon first use.
Comment 2: Line 26-31: Please specify the level of concentration.
Response 2: The specific concentrations used in the experiments have been clearly indicated.
Comment 3: Line 31-32: A clear path for further research should be suggested in the conclusion.
Response 3: The perspectives for future inestigations have been outlined in the Conclusion section.
Comment 4: For clarity, the introduction's first lengthy paragraph should be broken up into smaller ones. Line 64 might be shown as the beginning of a new paragraph.
Response 4: The long initial paragraph of the Introduction has been divided for better readability, beginning a new paragraph at Line 64.
Comment 5: Line 64-78: Numerous statements are provided without references.
Response 5: Appropriate references have been included to support the statements.
Comment 6: The molecular biology of colorectal cancer should be clearly reviewed in the introduction.
Response 6: The Introduction has been revised to provide context on the molecular biology of colorectal cancer.
Comment 7: The second paragraph in introduction is unclear and should be expand for clarity. For example, polyphenols in Line 80 and phytochemicals in Line 82 & 85 should be defined. Define chemotherapeutic drugs in Line 83-84 & 86.
Response 7: Definitions and examples have been added for clarity.
Comment 8: If the introduction discussed the molecular role of essential oils derived from different medicinal plants and the range of pathways impacted in colorectal cancer lines, that would be interesting.
Response 8: We expanded the Introduction to discuss the bioactivity of essential oils from medicinal plants and their mechanisms of action.
Comment 9: I would recommend adding a concluding paragraph to the introduction to explain the purpose of the paper in relation to what is currently understood by in vivo/vitro studies, thereby emphasizing the study's added value.
Response 9: A final paragraph has been added to summarize the study’s rationale and highlight its relevance in the context of in vivo/vitro research.
Comment 10: Each of the results and the discussion should be presented in its own section.
Response 10: In the revised version of the manuscript, the Results and Discussion have been separated the into individual sections, as recommended by the reviewer.
Comment 11: The discussion provides insufficient interpretation of the results. The section should be prepared by organizing information according to: the main findings and comparison of these findings with those reported in the literature, the strengths and weaknesses of the study and in relation to other studies, information about the present analyses and the implications of this study and future research directions.
Response 11: We have revised the Discussion to include a broader interpretation of the results and comparisons with existing literature, highlighting both the strengths and limitations of the study. The implications of the study and perspectives for further investigations have also been discussed.
Comment 12: Please update as many of the references were outdated (Ref # 15-19).
Response 12: Outdated references have been replaced with more recent and relevant studies.
Comment 13: Modest editing for language is required throughout the paper by a native English speaker.
Response 13: The English spelling, grammar and punctuation have been carefully checked and all noted errors were corrected.
Round 2
Reviewer 3 Report
Comments and Suggestions for Authors
Only one comment left to improve the paper.
Please specify the limitations of the study in one paragraph at the end of discussion.
Author Response
Comment 1: Please specify the limitations of the study in one paragraph at the end of discussion.
Response 1: Thank you for the suggestion. At the end of the Discussion section, we have added a paragraph that highlights the limitations of our research at this stage.